# Assessing Sensory Attributes and Properties of Infant Formula Milk Powder Driving Consumers’ Preference

**DOI:** 10.3390/foods12050997

**Published:** 2023-02-26

**Authors:** Yanmei Xi, Tong Zhao, Ruirui Liu, Fuhang Song, Jianjun Deng, Nasi Ai

**Affiliations:** 1Beijing Advanced Innovation Center for Food Nutrition and Human Health, School of Food and Health, Beijing Technology & Business University, Beijing 100048, China; 2State Key Laboratory of Vegetable Biobreeding, Institute of Vegetables and Flowers, Chinese Academy of Agricultural Sciences, Beijing 100081, China

**Keywords:** IFMP, sensory analysis, K-means, internal preference mapping

## Abstract

Infant formula milk powder (IFMP) is an excellent substitute for breast milk. It is known that the composition of maternal food during pregnancy and lactation and exposure level to food during infancy highly influence taste development in early infancy. However, little is known about the sensory aspects of infant formula. Herein, the sensory characteristics of 14 brands of infant formula segment 1 marketed in China were evaluated, and differences in preferences for IFMPs were determined. Descriptive sensory analysis was performed by well-trained panelists to determine the sensory characteristics of evaluated IFMPs. The brands S1 and S3 had significantly lower astringency and fishy flavor compared to the other brands. Moreover, it was found that S6, S7 and S12 had lower milk flavor scores but higher butter scores. Furthermore, internal preference mapping revealed that the attributes fatty flavor, aftertaste, saltiness, astringency, fishy flavor and sourness negatively contributed to consumer preference in all three clusters. Considering that the majority of consumers prefer milk powders rich in aroma, sweet and steamed flavors, these attributes could be considered for enhancement by the food industry.

## 1. Introduction

Breastfeeding is the best strategy for feeding newborn babies and provides a balanced supply of nutrients to ensure absorption and supports optimal infant growth and development. With advances in science and technology, the current understanding on the immunological, hormonal and nutritional properties of breast milk and how its composition can be tailored to the infant’s unique needs and environment has increased [1]. However, insufficient or lack of provision of breast milk may occur due to personal and social factors. Compared to other complementary foods, in the early stages of infant’s incomplete digestion and absorption physiology, IFMP is often used as a substitute for breast milk to meet the infant’s special nutritional needs while also reducing the stress of nursing mothers [2]. Currently, the demand for infant formula has been increasing in the Chinese market along with the gradual implementation of the national three-child policy. 

The sensory profile is the combination of sensory impressions which directly determine the consumer terminal and corporate image. Previous studies have found that the development of newborn’s gustatory functions during the typical forty-week gestation [3] is highly related to the food consumed by the mother during pregnancy as well as later during infancy and childhood [4,5]. Other studies revealed that baby facial expressions can be quantified to determine the development of the infant's taste preferences, which showed that taste development during infant’s early development is closely related to the composition of foods consumed during pregnancy and lactation as well as to the degree of food contact in infants and the ability to receive food during childhood [6,7]. In addition, it was found that taste receptivity of adult rats that were exposed to a certain taste during early postnatal development increased with age, which influenced subsequent physiology and behavior [8,9]. Moreover, infants lack the ability to discriminate between actual breast milk and infant formula since their language function is not perfect; hence, prospective parents are particularly concerned about the taste of infant formula. However, current research is primarily concerned with nutrition and quality control of milk powder [10,11,12,13,14]. Masum et al. [15] comprehensively and critically reviewed the available information on the effects of product formulation, processing and storage processes on the physicochemical properties of end-of-life infant formulae. Phosanam et al. [16] established correlations between storage environment, lactose crystallization, surface components of IMF powders and their agglomeration (including agglomeration intensity) to extend the shelf life of infant formula milk powder. Gredilla et al. [17] propose a rapid, simple and conclusive analytical method based on chemometrics for assessing the toxicity of hazardous elements in commercial infant formula. In our research, we found that consumers pay more attention to the sensory quality of infant formula when buying infant formula. Only a few studies have considered the sensory quality of this product. 

Therefore, the aim of this study was to investigate the sensory attributes of fourteen popular brands of powdered milk currently found in the Chinese market by a sensory panel. In addition, the factors affecting buying behavior, descriptive sensory analysis and consumer testing of powdered milk were analyzed by multiple statistical analyses. Moreover, internal preference mapping was used to analyze the factors affecting buying of infant milk powder in terms of sensory preferences. The results discussed herein provide insights into the sensory attributes of milk powder marketed in China and highlight consumer’s preferences as well as elucidate the preliminary sensory factors that block consumer acceptance of milk powder, thereby serving as a theoretical guide for both the manufacture and consumer choice related to milk powder. Distinct from previous studies, this paper highly focuses on the flavor quality of infant formula to promote the “flavor and health-oriented” development of the food industry.

## 2. Materials and Methods

### 2.1. Sample Collection

IFMPs from 14 popular brands currently available in the Chinese market under various trademark names were purchased at different Yonghui supermarket branches (Beijing, China). These brands were selected for the extension of their market share in China as well as for their representation on a survey conducted in fifty mother–infant households for two months. Milk powder samples were purchased in bulk, numbered with a label within the range of S1–S14, and sent to the dairy flavor laboratory at the Beijing Technology and Business University for subsequent analysis. Milk powder from the following regions was evaluated herein: S1 and S2 samples came from the Heilongjiang region, while S3, S4 and S5 were produced in the Inner Mongolia region. The Beijing region produced S6 and S7, sample S8 was from Zhejiang, Guangdong company provided S9, S10, S11 and S12, while S13 and S14 originated from Hong Kong and Taiwan, respectively. Basic information was collected on the 14 samples (Table 1), including sample name, protein, fat and carbohydrate percentage.

### 2.2. Screening of Sensory Panelists

Participants (aged 23–35 years) were enrolled in sensory analysis tests based on their ability to judge olfactory attributes, sensitivity to the evaluated attributes and verbal description. Participants were from the Dairy Flavor Chemistry Lab and had extensive experience in evaluating milk powder. The sensory evaluators are in good health, equipped with the ability to focus and remain free from outside influences. The candidate demonstrates interest and motivation in the sensory analysis and is able to attend sensory evaluations on time. In total, 56 panelists (45 females and 11 males) were recruited for further guidance and learning, who voluntarily participated in sensory evaluation of milk powder samples, and each panelist received an individual identification label.

### 2.3. Training of Sensory Panelists

Sensory panelists were trained to improve their ability to perceive, identify and describe the sensory attributes of milk powder [18]. Panelists were trained for 3 h each morning for 14 days. In total, nine sensory attributes were identified at the end of training, which included milk flavor, sweetness, aftertaste, saltiness, butter taste, astringent taste, fishy taste, sourness and steamed flavor. All samples were evaluated in two independent replicates, and three sensory attributes (i.e., butter taste, astringent taste, and aftertaste) during training were submitted to repeatability tests.

### 2.4. Descriptive Sensory Analysis

Descriptive sensory analysis included the analysis of aroma and taste of infant formula milk powder. Descriptive sensory evaluation criteria are shown in Table 2. Sensory evaluation applied the principles of experimental design and statistical analysis to the use of human senses, thus aiming to isolate the sensory properties of foods and to provide information on sensory properties of food products [19]. Formal sensory evaluation was conducted in a dairy flavor sensory evaluation laboratory at room temperature (25–28 °C) and under bright and soft LED lighting. Each sensory evaluator had an identical dedicated space free of disturbances. Each sample was dissolved in water at 25 °C, poured into a 50 mL evaluation cup labeled with a three-digit random number, and presented to each evaluator in random order. After identifying the sensory attributes of milk powder samples, their intensity was scored in a subsequent sensory evaluation [20,21]. Participants had free sensory evaluation preference tests that were conducted based on a previous study [22] using a nine-point scale to assess the intensity of sensory attributes of milk powder samples, in which 1 indicated the absence of the attribute, 5 indicated a moderately strong intensity, and 9 indicated very strong intensity. Participants rated the aroma and taste of milk powder samples, and they undertook a short (2–3 min) break after each sample evaluation. Participants had free access to water and sugar-free soda crackers for palate cleaning during sensory evaluation rounds. Data from 56 participants were recorded and summarized after evaluation.

### 2.5. Statistical Analysis

Sensory evaluation data were analyzed with SPSS 16.0 (IBM Deutschland GmbH, Ehningen, Germany) for one-way ANOVA and significance analysis by Duncan’s multiple-range tests. In addition, principal component analysis (PCA) was applied to explain the relationship between sensory attributes and sample characteristics using PanelCheck (Version 1.4.2). Furthermore, hierarchical cluster analysis (HCA) was performed on sample groups with similar sensory attributes and analytical characteristics based on Ward’s linkage using Euclidean distances. Additionally, panelists were segmented via the k-means clustering method. To further understand consumer’s preferences, the k-means clustering method was combined with PCA for external preference mapping. Finally, consumer preference scores were correlated with attribute intensity rated for descriptive analysis.

## 3. Results and Discussion

### 3.1. Sensory Evaluation Skills Assessment

It is known that the results of sensory evaluation analysis are influenced by the evaluator’s assessment ability [24]. The results obtained herein from 56 sensory evaluators were recorded and imported into Panel Check for subsequent analysis. In the case of milk powder sensory data, it means that the MSE values for a total of three attributes (butter, astringency, and aftertaste) can be visualized in one graph. F values, mean squared error (MSE) values and *p* * MSE are usually used to assess the ability of sensory evaluators [25]. F-value is the ratio of the difference between sample groups to the difference within groups; in general, if there is a larger F value on sensory attributes, it indicates that the evaluator has a better discrimination level of the sample. The MSE value indicates within-group variance and provides a measure for the reproducibility of evaluators’ judgement; lower MSE values indicate better repeatability of evaluators’ judgement (Figure 1). All 56 sensory evaluators indicated low MSE values (MSE < 3.5) for three sensory attributes, which revealed good repeatability of results after a fourteen-day training, thus ensuring the accuracy of experimental results.

### 3.2. Descriptive Sensory Analysis of Milk Powder

Flavor is an important attribute which determines consumer’s acceptance and preference of infant formula milk powder. Moreover, it has been shown that sensory properties of food products have been currently driving consumer liking [26]. In addition, the manufacture and promotion of foods are intrinsically associated with sensory evaluation. Thus, descriptive sensory analysis was conducted to evaluate sensory properties of commercial milk powder, and the results were analyzed by ANOVA to determine significance of observed differences in sensory properties of descriptive sensory analysis. In total, the 56 trained panelists proposed nine sensory attributes to describe milk powder samples. The significance of differences in the nine sensory attributes—butter, sweetness, saltiness, sourness, aftertaste, astringent taste, milk flavor, steamed flavor and fishy flavor and preference—are shown in Table 3. Figure 2 depicts the results of quantitative descriptive analysis from the radar plot and evaluation scores. The results showed that the milk flavor and sweetness of sample S3 were more pronounced compared to other samples. However, no significant differences were observed for sourness. In addition, fishy flavor, butter and the astringent taste and sourness in samples M6 and M7 were significantly higher than others in samples (*p* < 0.05), which may directly affect the evaluator’s preference. Interestingly, M6 and M7 were not significantly different from sample M12 in milk flavor scores, but they were significantly lower compared to the other milk powder samples (*p* < 0.05). Moreover, milk flavor scores were higher in samples S1, S3, S8 and S14 compared to other samples, which may resulted in higher preference.

The sensory histogram of aroma and taste is shown in Figure 3. Steamed, fishy and astringent flavors were the main sources of off-flavors in milk powder samples, and butter was the most important flavor precursor of oxidized flavor. Lipids are the most important flavor precursors of oxidized flavors in dairy powders, and milk fat oxidation can lead to off-flavors during the processing and storage of dairy powders. Oxidized flavors can be described as fatty, greasy, soapy, and dyestuff [27]. Hexanal and 2-heptanone are typical oxidative volatiles, which accelerate fat oxidation during preheating, concentration, drying and storage at 40 °C [28]. Hexane, heptane and pentane correlated with “painty”, “oxidized”, “cooked”, and “caramelized” attributes in dairy based powders during storage [29]. In addition, steamed flavor is described as accompanied by a caramel fragrance. Lactose in cow’s milk may undergo degradation, Maillard reaction and isomerization, which can lead to deepening its color as well as the generation of furanone, volatile furanones, furfural and malt phenols, which contribute to the development of steamed and caramel flavor in milk powder during processing or long-term storage [30]. The heating of milk powders can trigger a Maillard reaction, producing furfural, furan derivatives, dicarbonyl compounds and other flavor substances, 2-ethylfuran, which produced the caramel aroma of milk powders [31]. Furthermore, the endogenous enzymes found in cow’s milk can degrade proteins and lead to the production of flavor precursors such as taste peptides and amino acids with sweet and sour flavors such as bitterness and astringency. The astringency is attributed to the interaction product of whey protein, calcium phosphate, and caseins during the high-temperature treatment [32], which are attributes that limit consumer’s choice. Interestingly, The samples scored differently in terms of saltiness, and process variation may lead to differences in the degree of lactose degradation, which may contribute to differences in flavor.

### 3.3. Relationship between Sensory Attributes and Quality Grades of Milk Powder

ANOVA failed to identify the interaction between sensory descriptors and the overall sensory characteristics and their degree of variation in milk powder samples. Thus, sensory data were further analyzed by PCA. 

Standardized PCA plots using correlation matrices are commonly used in sensory analysis to visualize the relationships between sensory properties, individual samples and the strength of correlation between samples and sensory attributes [33]. PCA is a classification technique which consists of eigenvectors coupled with a correlation matrix, in which the maximum difference between data can be revealed by axes rotation. A new set of axes is then calculated by dimensionality reduction to capture the maximum difference between the entire data set. Thus, PCA enables data visualization by reducing the dimensionality of complex datasets, which increases interpretability and minimizes information loss [21]. Pan [34] compared the differences among skim milk samples processed under different preheating treatments by combining E-tongue with PCA and cluster analysis (CA). Moreover, Chi et al. [35] observed variation trends both in volatile favor composition and aroma release in six skim milk products using PCA. Hence, PCA showed the relationship between the classification contributions of different milk powder brands and the internal association between samples and sensory attributes.

Figure 4 shows the PCA plot, which revealed that the contribution of PC1 was 73.2%, whereas the contribution of PC2 was 9.2%, and these two dimensions described 82.4% of the variability in the data set. In the first dimension, samples S2, S6, S7, S9, S12, and S13 were clearly separated from samples S1, S3, S4, S5, S8, S10, S11 and S14. The first dimension (F1) enabled distinguishing milk powder samples by the intensity of the following attributes: sweetness, milk flavor, steamed flavor and preference. Conversely, the attributes associated with poorer milk powder quality were butter taste, fishy flavor, sourness, astringency, saltiness and aftertaste, which were found on the positive side of the plot. Furthermore, samples S6, S7, and S12 were strongly correlated with less favorable attributes such as saltiness, butter taste, fishy flavor and astringency. Conversely, milk flavor and sweetness contributed greatly to sensory descriptions of samples S3 and S1 in the first dimension, being thus preferred by sensory evaluators. Finally, the first dimension enabled distinguishing samples S6, S7, S12, S9, S13, and S2 from other samples. 

Moreover, CA was further used to analyze 14 samples based on the evaluation results of sensory evaluators. CA graph showed that samples S6, S7 and S12 clustered together, which revealed that evaluators were less receptive to these samples based on sensory attributes and preference analysis. Taken together, the PCA (Figure 4a), CA (Figure 4b) and sensory analysis (Figure 2) results were consistent and fully reflected the negative evaluation of sensory attributes of samples S6, S7, and S12.

### 3.4. Preference Analysis and External Preference Mapping

K-means is an unsupervised clustering algorithm that allows deciding a priori on the number of cluster groups. In a previous study, extra virgin olive oil was classified into two categories based on the k-means algorithm [36]. K-means clustering is a common statistical data analysis that captures consumer preferences in detail and accounts for the diversity of evaluators’ preferences. CA enables the selection of three preference groups, with different groups of consumers with different preferences for milk powder samples [37].

Figure 5 shows the results of ratings of milk powder samples for each preference group. The first preference group consisted of 17 members that contained more than five points for two samples, (i.e., S1, S3 and S11); members of this group had an average preference for most milk powder samples with an average preference score of 4.08, and this group was composed of female members. Hence, it can be considered that women have a lower preference for the milk powder samples evaluated herein. In addition, cluster 2 (n = 23) had higher preference scores for all milk powder samples compared to the members of cluster 1 and cluster 3. The average score was 5.11. Thus, members of this group showed a relatively high approval of the most milk powder samples. The preference for samples M6, M7, and M12 was below 4, which was consistent with the other cluster. Finally, the third preference group consisted of 16 members, which rated preference for samples S1, S2, S3, S4 and S5 above 5. Meanwhile, the scores of M6, M7, M12 were 2.6, 2.8, and 2.32, respectively. All groups have poor sensory impressions of M6, M7 and M12.

Then, to further investigate the influence of personal liking on the sensory description of milk powder samples, PCA factor scores were linked to CA results to conduct external preference mapping (in Figure 6). Cluster 1 (n = 17) and cluster 2 (n = 23) had similar preferences, since milk sweetness was appointed as an essential trait, which were distinct from cluster 3 (n = 16) when considering preference. Interestingly, cluster 1 scored below 4 for five milk powder samples. Moreover, most of the consumers (n = 23) associated milk powder flavor with the attribute sweetness, milk flavor and steamed flavor (S3 and S4). However, as identified in the present study, consumers tend to dislike saltiness, butter taste, astringency, fishy flavor, and aftertaste in milk powder. Therefore, different sensory attributes affect the degree of consumer preference for milk powder and may also affect brand choice. Moreover, cluster 3 was associated with milk flavor and cooking flavor in quadrant 3, since a higher consumers’ acceptance was observed for the majority of samples. Cluster 3 reported preference for milk flavor, steamed flavor and preference scores above 4 for seven samples (Table 3). In addition, all clusters were located on the negative half-axis of PC1, which enabled distinction between samples S6, S7 and S12, which had fatty flavor, saltiness, fishy flavor, astringency and aftertaste. Thus, these odor attributes (i.e., fatty flavor, saltiness, fishy flavor, astringency and aftertaste) hindered consumers’ choice of milk powder. Moreover, two sources of origin and milk powder brands, i.e., S6 and S7, were derived from Beijing, and S12 was obtained from Guangdong, and differences in origin and milk powder brand may be an important factor affecting the sensory quality and preference of milk powder.

## 4. Conclusions

To date, no studies have been conducted on Chinese milk powder samples for specific sensory evaluation and preference descriptions. Herein, descriptive sensory analyses were conducted on fourteen milk powder samples currently available in the Chinese market. Nine sensory descriptors were outlined to describe the overall sensory characteristics and preferences of evaluated milk powder samples, which enabled a more valid and accurate assessment. In addition, k-means enabled classifying the sensory evaluators into three clusters based on preference. Moreover, external preference mapping showed that the three clusters were clearly separated from milk powder samples associated with saltiness, aftertaste, creamy flavor, astringent flavor, cow flavor, and sourness. Milk powder samples with a predominance of milk sweetness, milk flavor and steamed flavor were the preferred sensory attributes, which point toward the attributes that should be pursued by the industry. Finally, an in-depth understanding of flavor formation in IFMPs might improve the processing and quality of milk powder to reduce the formation of off-flavors.

## Figures and Tables

**Figure 1 foods-12-00997-f001:**
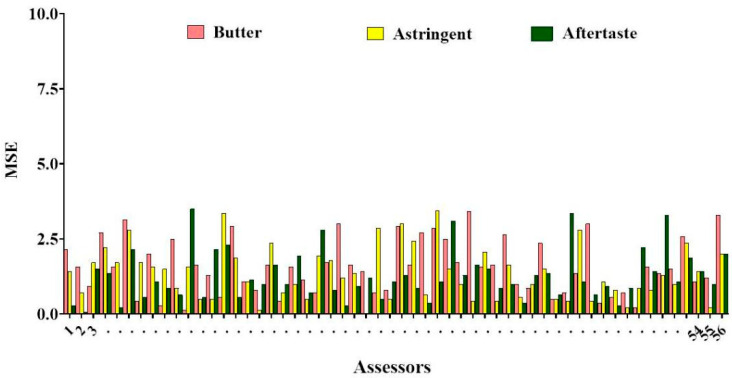
Analysis of assessment capacity of sensory evaluators enrolled in this study of fourteen Chinese milk powder samples. Mean squared error (MSE) values indicate within-group variance of sensory evaluators. 1, 2, 3... 56 represent 56 sensory evaluators.

**Figure 2 foods-12-00997-f002:**
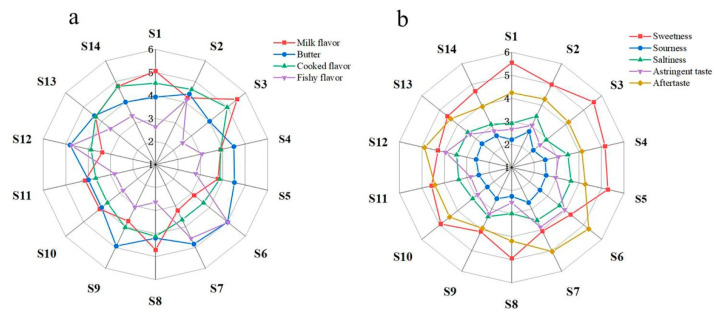
Radar chart of the sensory evaluation of fourteen milk powder samples available in the Chinese market. Samples were individually assigned a label within the range of S1–S14. (**a**) Represent odor score, (**b**) represents taste score.

**Figure 3 foods-12-00997-f003:**
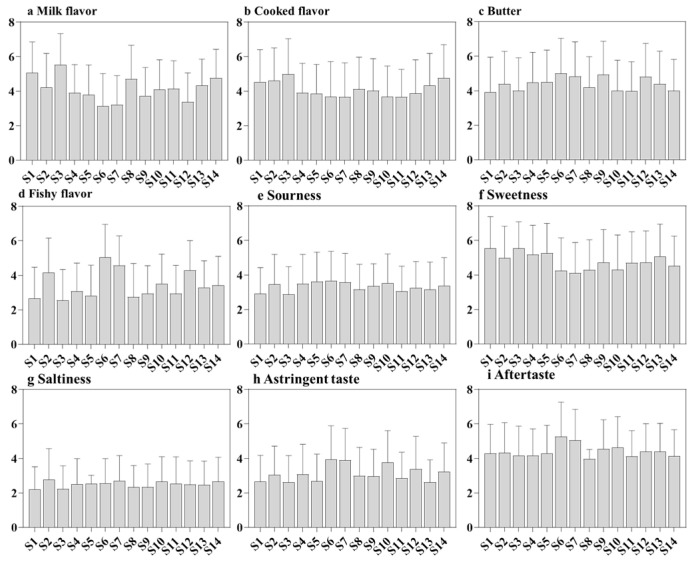
Histogram of odor and taste attributes rating of fourteen milk powder samples (labeled individually from S1 to S14) available in the Chinese market. (**a**–**d**) Histograms of odor sensory scores. (**e**–**i**) Histograms of taste scores.

**Figure 4 foods-12-00997-f004:**
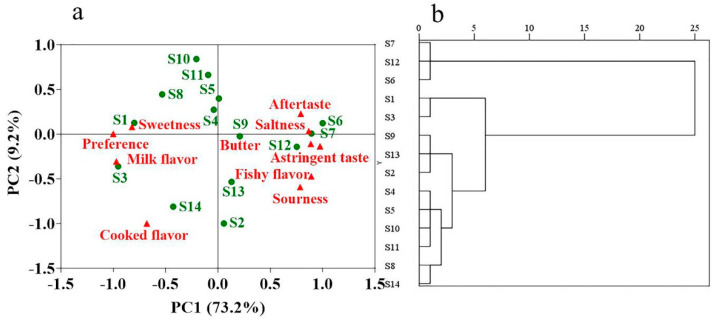
Classification of milk power samples from different brands currently marketed in China. (**a**) Biplots based on the results of principal component analysis from the descriptive analysis of fourteen milk powder samples. The first two dimensions of the biplot explained 82.4% of the variation among the samples. (**b**) Dendrogram of cluster analysis. Graphical representation of samples from consumer preference on fourteen infant formula milk powder samples.

**Figure 5 foods-12-00997-f005:**
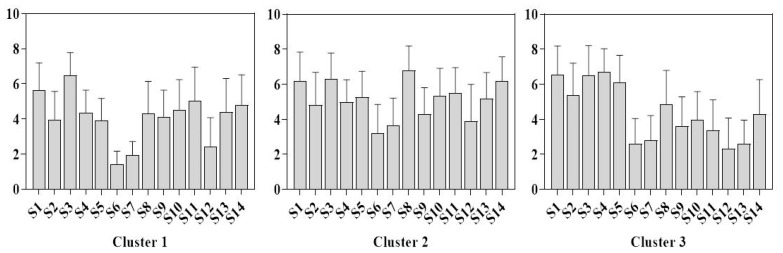
K-means clustering analysis of preference scores for fourteen milk powder samples currently available in the Chinese market.

**Figure 6 foods-12-00997-f006:**
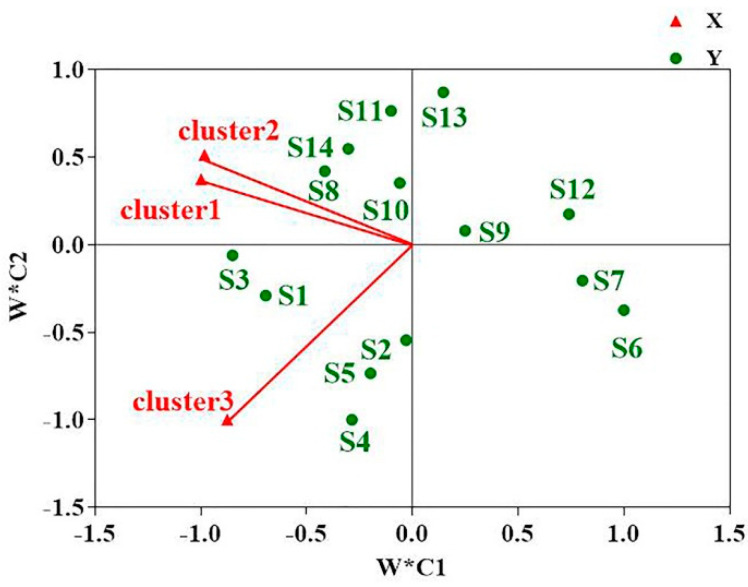
External preference mapping based on results of principal component analysis and agglomerative hierarchical clustering analysis. Cluster1, cluster2, and cluster3 refer to the three cluster groups formed based on K-means clustering analysis of preference, respectively.

**Table 1 foods-12-00997-t001:** Sample Information.

Sample	Fat (Per 100 g Milk Powder)	Protein (Per 100 g of Milk Powder)	Carbohydrates (Per 100 g of Milk Powder)
S1	25	11	55.2
S2	27	10.9	54.4
S3	28	10.8	53.2
S4	24	10.6	51.6
S5	28.1	10.2	50.2
S6	27	12	56
S7	26.1	11.7	56.4
S8	26.04	11.99	51.9
S9	26.8	11.5	54.6
S10	26.6	11.6	56.1
S11	25	11	55.2
S12	27	11.5	56.5
S13	27.9	11.1	52.8
S14	25	11.8	54.4

**Table 2 foods-12-00997-t002:** Sensory attributes and standard of sensory evaluation [23].

No.	Attributes	Evaluation Method
1	Milk flavor	Vanillin added to milk powder base, inherent flavor of milk powder, the aroma is gentle and scented
2	Milk fat	E, E-2,4-decadienal added to milk powder bases, intensity of greasy feeling in the mouth, harsh taste intensity of oil oxidation
3	Cooked	Milk powder substrate in water bath at 85 °C for 30 min, steaming aroma on the nose, with a slight caramel aroma
4	Fishy flavor	Freshly slice of mushrooms are added to the milk powder base, smell a metallic, mud-like fishy smell
5	Sweet taste	5% sucrose solution, intensity of the inherent sweetness for sample
6	Sour	1% citric acid solution, intensity of sourness was felt in the mouth
7	Saltiness	2% NaCI solution, intensity of salty taste perceived by the mouth
8	Astringent	Soak 6 tea bags in water for 10 min, the intensity of an astringent sensation felt by the tongue
9	Aftertaste	Sensory intensity of residual milk flavor, softness and fluidity remaining in the mouth and throat
10	Fullness	Overall, degree of preference for test samples

**Table 3 foods-12-00997-t003:** Significance analysis of sensory evaluation results (mean and standard deviation, SD of milk powder samples).

Sample	Milk Flavor ^#,†,‡^	Milk Fat	Cooked	Fishy Flavor	Sweet Taste	Saltiness	Sour	Astringent	Aftertaste	Preference
S1	5.10 ± 1.80 ^ab^	3.99 ± 2.03 ^d^	4.48 ± 1.89 ^abcd^	2.67 ± 1.65 ^e^	5.54 ± 1.84 ^a^	2.21 ± 1.31 ^c^	2.91 ± 1.52 ^bc^	2.65 ± 1.53 ^d^	4.29 ± 1.68 ^cde^	6.03 ± 1.66 ^a^
S2	4.24 ± 1.99 ^d^	4.35 ± 1.92 ^cd^	4.61 ± 1.91 ^abc^	4.17 ± 2.21 ^b^	4.99 ± 1.84 ^bc^	2.76 ± 1.81 ^a^	3.46 ± 1.73 ^abc^	3.04 ± 1.67 ^cd^	4.33 ± 1.74 ^cde^	4.73 ± 1.86 ^cd^
S3	5.55 ± 1.79 ^a^	3.99 ± 1.94 ^d^	5 ± 2.09 ^a^	2.55 ± 1.77 ^e^	5.54 ± 1.55 ^a^	2.22 ± 1.35 ^bc^	2.88 ± 1.60 ^a^	2.62 ± 1.54 ^d^	4.17 ± 1.70 ^cde^	6.4 ± 1.46 ^a^
S4	3.92 ± 1.63 ^de^	4.46 ± 1.77 ^abcd^	3.88 ± 1.72 ^ef^	3.08 ± 1.85 ^cde^	5.18 ± 1.70 ^ab^	2.5 ± 1.49 ^abc^	3.49 ± 1.70 ^abc^	3.08 ± 1.74 ^cd^	4.17 ± 1.54 ^cde^	5.14 ± 1.51 ^bc^
S5	3.79 ± 1.77 ^def^	4.54 ± 1.89 ^abcd^	3.82 ± 1.75 ^ef^	2.82 ± 1.53 ^de^	5.26 ± 1.73 ^ab^	2.52 ±0.50 ^abc^	3.62 ± 1.77 ^ab^	2.68 ± 1.57 ^cd^	4.3 ± 1.62 ^cde^	4.95 ± 1.79 ^c^
S6	3.16 ± 1.92 ^h^	5.02 ± 2.08 ^a^	3.65 ± 2.06 ^f^	5.04 ± 2.55 ^a^	4.25 ± 1.90 ^de^	2.56 ± 1.43 ^abc^	3.67 ± 1.71 ^a^	3.95 ± 1.94 ^a^	5.27 ± 1.99 ^a^	2.52 ± 1.74 ^f^
S7	3.24 ± 1.71 ^gh^	4.85 ± 1.04 ^abc^	3.66 ± 1.98 ^f^	4.58 ± 2.46 ^ab^	4.11 ± 1.77 ^e^	2.71 ± 1.45 ^a^	3.57 ± 1.68 ^ab^	3.89 ± 1.85 ^a^	5.04 ± 1.81 ^ab^	2.88 ± 1.55 ^f^
S8	4.74 ± 1.94 ^cd^	4.18 ± 1.81 ^d^	4.15 ± 1.86 ^cdef^	2.75 ± 1.79 ^de^	4.29 ± 1.75 ^de^	2.33 ± 1.26 ^abc^	3.16 ± 1.46 ^bcde^	2.99 ± 1.65 ^cd^	3.97 ± 0.55 ^e^	5.5 ± 2.03 ^b^
S9	3.69 ± 1.61 ^efg^	4.96 ± 1.96 ^ab^	4.04 ± 1.88 ^def^	2.95 ± 1.85 ^cd^	4.73 ± 1.91 ^bcd^	2.33 ± 1.35 ^abc^	3.35 ± 1.30 ^abde^	2.97 ± 1.56 ^d^	4.55 ± 1.69 ^cd^	4.12 ± 1.55 ^e^
S10	4.14 ± 1.71 ^de^	4.01 ± 1.82 ^d^	3.71 ± 1.81 ^f^	3.52 ± 2.11 ^c^	4.31 ± 2.01 ^de^	2.65 ± 1.43 ^abc^	3.54 ± 1.68 ^abc^	3.76 ± 1.85 ^ab^	4.64 ± 1.78 ^bc^	4.73 ± 1.70 ^cd^
S11	4.15 ± 1.64 ^de^	3.98 ± 1.72 ^d^	3.69 ± 1.62 ^f^	2.94 ± 1.65 ^cde^	4.71 ± 1.80 ^bcd^	2.54 ± 1.55 ^abc^	3.06 ± 1.45 ^cde^	2.86 ± 1.50 ^cd^	4.11 ± 1.50 ^de^	4.88 ± 1.89 ^c^
S12	3.33 ± 1.71 ^fgh^	4.83 ± 1.95 ^abc^	3.88 ± 1.97 ^ef^	4.3 ± 2.16 ^b^	4.73 ± 1.82 ^bcd^	2.49 ± 1.37 ^abc^	3.24 ± 1.54 ^abcde^	3.37 ± 1.91 ^bc^	4.39 ± 1.62 ^cde^	2.99 ± 1.99 ^f^
S13	4.3 ± 1.54 ^d^	4.43 ± 1.93 ^bcd^	4.34 ± 1.91 ^bcde^	3.3 ± 1.74 ^cd^	5.08 ± 1.87 ^ab^	2.46 ± 1.39 ^abc^	3.15 ± 1.60 ^bcde^	2.62 ± 1.30 ^d^	4.39 ± 1.64 ^cde^	4.28 ± 1.87 ^de^
S14	4.75 ± 1.68 ^bc^	4.02 ± 1.83 ^d^	4.78 ± 1.97 ^ab^	3.42 ± 1.94 ^c^	4.52 ± 1.74 ^cde^	2.66 ± 1.41 ^ab^	3.39 ± 1.62 ^abcd^	3.23 ± 1.67 ^c^	4.13 ± 1.53 ^cde^	5.21 ± 1.83 ^bc^

^#^ n = 56, ^†^ 9-point scale 1 = absence of the attribute, 5 = moderately strong for an attribute, 9 = very strong for an attribute. ^‡^ Means in each column with the same letter are not significantly difference (*p* > 0.05).

## Data Availability

Data is contained within the article.

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
