# Peer review of "Assessing Sensory Attributes and Properties of Infant Formula Milk Powder Driving Consumers’ Preference"

_foods, 2023, doi:10.3390/foods12050997_

Round 1

Reviewer 1 Report

Interesting work. Please find my coments in attached file.

Reviewer 2 Report

This is an interesting and well-researched topic. While the paper is easy to read; however, several issues diminish its importance and contribution.

The paper is methodologically well designed and congruent with the previous relevant theory and empirical reasons.

The results are well analyzed and discussed well linked to the findings of past studies. However, the novelty of this study should also discuss within the scope of world literature.

Theoretical and practical implications of the research results as well as scientific contribution of the paper are not sufficiently elaborated. What is the paper's contribution to theory and food industry of infant formula milk powder?

Reviewer 3 Report

This manuscript evaluated the sensory characteristics of infant formula milk powder from 14 different brands by 56 trained panelists. The topic is interesting and the manuscript is well-designed. It can be accepted after a minor revision.

1. It is better to change the article type to a Brief Report or a Case Report.

2. L 74; Please label each brand for better understanding; S1, S2, etc. Also, more details about them are needed, for example manufacturing company, country, etc.

3. Figure 1 is unclear. Please re-plot it.

4. Figures 3 and 5; Check the standard deviations and add the significant letters.

5. For a better conclusion, it would have been better to provide another parameter, i.e. overall acceptability, in the sensory characteristics section.
